# Surrogate Gradient Design for LIF networks

## Abstract

Spiking Neuromorphic Computing uses binary activity to improve Artificial Intelligence energy efficiency. However, the non-smoothness of binary activity requires approximate gradients, known as Surrogate Gradients (SG), to close the performance gap with Deep Learning. Several SG have been proposed in the literature, but it remains unclear how to determine the best SG for a given task and network. Good performance can be achieved with most SG shapes, after a costly search of hyper-parameters. Thus, we aim at experimentally and theoretically define the best SG across different stress tests, to reduce future need of grid search. Here we first show that more complex tasks and network need more careful choice of SG, and that overall the derivative of the fast sigmoid outperforms other SG across tasks and networks, for a wide range of learning rates. Secondly, we focus on the Leaky Integrate and Fire (LIF) spiking neural model, and we note that high initial firing rates, combined with a sparsity encouraging loss term, can lead to better generalization, depending on the SG shape. Finally, we provide a theoretical solution, inspired by Glorot and He initializations, to find a SG and initialization that experimentally result in improved accuracy. We show how it can be used to reduce the need of extensive grid-search of dampening, sharpness and tail-fatness.

## 1 Introduction

Spiking Neuromorphic Computing uses binary and sparse signals to construct learning algorithms with higher energy efficiency (Henderson et al., 2020; Blouw et al., 2019; Davies et al., 2021; Lapique, 1907; Izhikevich, 2003). However, a binary signal means that the true derivative is zero essentially always, and training with gradient descent will be at best very poor. Research has shown that designing an approximate gradient, referred to as Surrogate Gradient (SG) (Esser et al., 2016; Zenke and Ganguli, 2018; Bellec et al., 2018), significantly improves training success. However, that entails an additional hyper-parameter to choose: the SG to use. Additionally, the best SG can depend on the neural architecture chosen, on the task, on the learning rate, on the initialization, and so on, making it difficult to know a priori which to pick. Thus, finding the best SG for a particular setting, requires a time consuming grid search, and reducing that search time is desirable.

To meet that need, we stress test a wide variety of SG, focusing on one specific neuron model, the Leaky Integrate and Fire (LIF) (Lapique, 1907; Gerstner et al., 2014), and provide a mathematical solution based on gradient stability methods (Glorot and Bengio, 2010; He et al., 2015), to design the best SG for a LIF. In contrast, it is standard to pick one SG for all the experiments (Bohte, 2011; Hubara et al., 2016a; Bellec et al., 2018; Zenke and Ganguli, 2018; Zenke and Vogels, 2021; Yin et al., 2021), possibly exploring the effect of changing a width factor (sharpness) (Zenke and Ganguli, 2018) or a height factor (dampening) (Bellec et al., 2018). Only in the past five years, the possibility of choosing an optimal SG has been considered (Neftci et al., 2019; Zenke and Vogels, 2021). Moreover, even if many SG can achieve good performance (Zenke and Vogels, 2021), some shapes have more chances to fail training or achieve lower accuracy. It seems therefore valuable to have a complete picture of when and where each SG works, and which ones are better left behind.

For example, on more complex neural models and tasks, we measure an increase in sensitivity to the choice of SG with complexity, and observe some to degrade more gracefully, which stresses the need to pick the right SG in each setting. We then focus on arguably the simplest spiking neural model, the LIF, and confirm that the initialization scheme has different impact on each SG. Finally, to be able to propose our theoretical solution, we need to justify the use of high firing rates. Fortunately, we observe that low initial sparsity, can help generalization with high final sparsity. We use this observation to better justify, that setting the network on a high firing rate at the beginning of training, is not in contrast with a low firing rate at the end of training. Taking this finding into consideration, and in the spirit of Glorot and He initializations (Glorot and Bengio, 2010; He et al., 2015), we propose four conditions that keep the representations and gradients stable with time. We show that these conditions provide hyper-parameters that result in improved performance without additional hyper-parameter grid-search. When we observe closely the fine details of the SG shape, such as (1) its dampening, (2) its sharpness, and (3) how fast it decays to zero, i.e. tail-fatness, we see that the theoretically justified choice tends to be close to the best experimental choice.

Our contribution is therefore

- We show how task and network complexity, lead learning to be more sensitive to the choice of SG;
- We observe that the derivative of the fast-sigmoid outperforms other SG across tasks and networks;
- High initial firing rate can promote generalization with low final firing rate;
- We provide a theoretical method for SG choice based on bounding representations that improves experimental performance.
- Our method predicts dampening, sharpness and tail-fatness, that lead to high accuracy experimentally on the LIF network;

## 2 Preliminaries

### 2.1 Initialization Schemes

Our theoretical method for SG choice is based on techniques from the weights initialization literature, that we use in an unorthodox way, to design a SG. The initial values of the network parameters have a strong impact on training speed (Hanin and Rolnick, 2018) and peak performance (Glorot and Bengio, 2010; He et al., 2015). The theory often focuses on fully-connected feed-forward networks (FFN), given their mathematical tractability (Roberts et al., 2022). FFNs are defined as $\boldsymbol{y}_l = \boldsymbol{b}_l + W_l \sigma(\boldsymbol{y_{l-1}})$, where $\boldsymbol{y}_0$ is the data, $\boldsymbol{y}_L$ is the network output at depth $L$, $\sigma(\cdot)$ an activation and $\boldsymbol{b}_l \in \mathbb{R}^{n_l}, W_l \in \mathbb{R}^{n_l \times n_{l-1}}$ are the layer biases and weights, where $n_l$ is layer $l$ size. Typically, biases are sampled as zero and weights such that $Mean[W_l] = 0$ and $Var[W_l] = c_l$. The general recommendation is a $1/c_l \propto n_l$ to avoid exploding variance of representations (Glorot and Bengio, 2010; He et al., 2015). (Glorot and Bengio, 2010) finds $Var[W_l] = 2/(n_{l-1} + n_l)$ optimal for linear networks ($\sigma(y) = y$), known as *Glorot* initialization, while (He et al., 2015) finds $Var[W_l] = 2/n_{l-1}$ for *ReLU* networks ($\sigma(y) = max(0, y)$), known as *He* initialization. Instead, (Saxe et al., 2014) finds a column orthogonal $W_l$ optimal for linear networks, known as *Orthogonal* initialization. Usually $W_l$ elements are drawn from a uniform or a normal distribution. We propose the BiGamma distribution, such that $w_{ij} \sim \text{Gamma}(w; \alpha, \beta)/2 + \text{Gamma}(-w; \alpha, \beta)/2$, Fig. 1. The BiGamma keeps the optimal variance and orthogonality without sampling zeros.

On the contrary, theoretical justification for recurrent networks initialization has been proposed for the LSTM (Mehdipour Ghazi et al., 2019), and other non spiking recurrent networks (Hochreiter et al., 2001; Arjovsky et al., 2016; Pascanu et al., 2013). However, on spiking recurrent networks, an initialization theory is missing, since arguments such as the Echo State Network (Jaeger et al., 2007) do not apply to non convex activations, or activations without a slope one regime. In practice, (Zenke and Vogels, 2021) samples a $Var[W_l] = 1/3n_{l-1}$ Uniform, while (Bellec et al., 2018) a $Var[W_l] = 1/n_{l-1}$ Normal distribution, for similar spiking models.

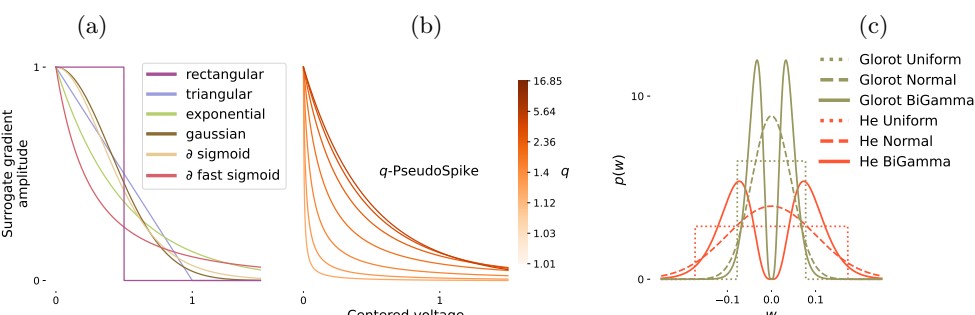

Figure 1: **Surrogate Gradient shapes and initialization distributions.** Panel (a) shows the SG investigated in this work, and (b) the tail dependence of the $q$-PseudoSpike SG for $q \in [1.01, 16.85]$. The SG considered are symmetrical around $v_t = y_t - \vartheta = 0$, so we only plot half the curve (centered voltage $v_t > 0$). Panel (c) shows the weight sampling distributions used, Gaussian (dashed), Uniform (dotted) and BiGamma (solid), with *He*, orange. and *Glorot* initialization, green, for a weight shape of $(n_0, n_1) = (200, 300)$.

## 2.2 Neural Models and Notation

Arguably the simplest spiking recurrent network model is the LIF (Lapique, 1907; Gerstner et al., 2014; Woźniak et al., 2020). It is defined as $\boldsymbol{y}_t = \boldsymbol{\alpha}_{decay}\boldsymbol{y}_{t-1}(1 - \boldsymbol{x}_{t-1}) + \boldsymbol{i}_t$ where $\boldsymbol{i}_t = W_{rec}\boldsymbol{x}_{t-1} + W_{in}\boldsymbol{z}_t + \boldsymbol{b}$, and $y_t$ is the neuron membrane voltage, using (Glorot and Bengio, 2010; He et al., 2015) notation. We define $x_t = \sigma(y_t) = \tilde{H}(y_t - \vartheta) = \tilde{H}(v_t)$ as the spiking activity, where $\vartheta$ is the spiking threshold, $v_t = y_t - \vartheta$ the centered voltage, and $\tilde{H}(v_t)$ a Heaviside function with SG. The term $(1 - \boldsymbol{x}_{t-1})$ represents a hard reset, that takes the voltage to zero after firing. The input $z_t$ can represent the data, or a layer below. It is common to write $\alpha_{decay} = 1 - \frac{dt}{\tau_m}$, where $dt$ is the computation time, $\tau_m$ the membrane time constant, and to multiply the other terms by biologically meaningful constants, that we compress for cleanliness. Each neuron can have its own speed $\alpha_{decay}$, intrinsic current $b$ and $\vartheta$. In this work, all the parameters in the LIF definition are learnable.

We denote vectors as $\boldsymbol{a}$, matrices as $A$, and their elements as $a$. The matrix $W_{rec} \in \mathbb{R}^{n_l \times n_l}$ connects neurons in the same layer, with zero diagonal, and $W_{in} \in \mathbb{R}^{n_l \times n_{l-1}}$ connects the layer with the layer below, or the data if $l = 0$. We use curved brackets $A(\cdot)$ for functions, and square brackets $A[\cdot]$ for functionals that depend on a probability distribution. We use interchangeably $\overline{x} = Mean[x]$, $\hat{x} = Max[x]$, and $\check{x} = Min[x]$. We use $\theta$ for any parameter. In a stack of layers, we add an index $l$ to each parameter and variable. Since the equation only depends on the previous timestep and layer, the probability distribution is a Markov chain in time and depth. Therefore the statistics we discuss are computed element-wise with respect to the distribution $p(y_{t,l}|t,l) = p(\boldsymbol{y}_{t-1,l}, \boldsymbol{z}_{t,l-1}, W_{rec,l}, W_{in,l}, b_l, \alpha_{decay,l}, \vartheta_l|t,l)$.

Additionally, we want to understand how the neuron complexity affects SG training. When a LIF is upgraded with a dynamical threshold to maintain longer memories, we have the Adaptive LIF (ALIF) (Gerstner et al., 2014; Bellec et al., 2018). Thus, we compare LIF to ALIF, and we propose the spiking LSTM (sLSTM), App. G, defined by changing the LSTM (Hochreiter and Schmidhuber, 1997) activations for neuromorphic counterparts.

## 2.3 Surrogate Gradients

As seen in the previous section, a spike is produced when the voltage surpasses the threshold, which mathematically can be described through a Heaviside function, $\tilde{H}(v)$, that is zero for $v < 0$ and one for $v \geq 0$. We use the tilde to remind that a SG is used for training, defined as $\tilde{H}'(v) = \gamma f(\beta \cdot v)$, where $\beta$ is the sharpness, $\gamma$ the dampening, $f$ is the shape of choice and $\cdot$ the product. Therefore $\gamma$ controls the maximal amplitude of the SG, and $\beta$ controls the width. A high sharpness, mostly passes the gradient for $v$ close to zero, while low sharpness also passes the gradient for a wider range of voltages. Therefore, the gradient can pass when the neuron has not fired. Unless explicitly stated, dampening and sharpness are set to one.

The SG shapes $f$ we investigated are (1) rectangular (Hubara et al., 2016b), (2) triangular (Esser et al., 2016; Bellec et al., 2018), (3) exponential (Shrestha and Orchard, 2018), (4) gaussian (Yin et al., 2020), (5) the derivative of a sigmoid (Zenke and Vogels, 2021), and (6) the derivative of a fast-sigmoid, also SuperSpike (Zenke and Ganguli, 2018). To make the comparison between different SG fair, $f$ is chosen to have a maximal value of 1 and an area under the curve of 1. We also propose a generalization of the derivative of the fast-sigmoid, that we call $q$-PseudoSpike SG. Its tail fatness is controlled by a hyper-parameter $q$ and we use it to study tail dependence in section 3.5. More in Fig. 1 and App. D.

## 2.4 Datasets

More details on the datasets can be found in App. A.

**Spike Latency MNIST (sl-MNIST):** the MNIST digits (LeCun et al., 1998) pixels (10 classes) are rescaled between zero and one, presented as a flat vector, and each vector value $x$ is transformed into a spike timing using the transformation $T(x) = \tau_{eff} \log(\frac{x}{x-\vartheta})$ for $x > \vartheta$ and $T(x) = \infty$ otherwise, with $\vartheta = 0.2, \tau_{eff} = 50$ms (Zenke and Vogels, 2021). The network input is a sequence of 50ms, 784 channels ($28 \times 28$), with one spike per row.

**Spiking Heidelberg Digits (SHD):** is based on the Heidelberg Digits (HD) audio dataset (Cramer et al., 2020) which comprises 20 classes of spoken digits, from zero to nine, in English and German, spoken by 12 individuals. These audio signals are encoded into spikes through an artificial model of the inner ear and parts of the ascending auditory pathway.

**PennTreeBank (PTB):** is a language modelling task. The PennTreeBank dataset (Marcus et al., 1993), is a large corpus of American English texts. We perform next time-step prediction at the word level. The vocabulary consists of 10K words, which we consider as 10K classes. The one hot encoding of words can be seen as a spiking representation, even if it is the standard representation in the non neuromorphic literature.

## 2.5 Training Details

Our networks comprise two recurrent layers. The output of each feeds the following, and the last one feeds a linear readout. Our LIF network has 128 neurons per layer on the sl-MNIST task, 256 on SHD, and one layer of 1700 and another of 300 on PTB, as in (Woźniak et al., 2020). On the SHD task, the ALIF has 256 neurons and the sLSTM 85, to keep a comparable number of 350K parameters. We train on the crossentropy loss. The optimizer had a strong effect, where Stochastic Gradient Descent (Robbins and Monro, 1951; Kiefer and Wolfowitz, 1952) was often not able to learn, and AdaM (Kingma and Ba, 2015) performed worse than AdaBelief (Zhuang et al., 2020). AdaBelief hyper-parameters are set to default, as in (Radford et al., 2018; Zenke and Vogels, 2021). The remaining hyper-parameters are reported in App. A. Unless explicitly stated, we use Glorot Uniform initialization. Each experiment is run 4 times and we report mean and standard deviation. Experiments are run in single Tesla V100 NVIDIA GPUs. We call our metric the mode accuracy: the network predicts the target at every timestep, and the chosen class is the one that fired the most for the longest.

## 3 Results

### 3.1 Sensitivity to Complexity

#### 3.1.1 Methodology

In order to portray the difficulty of choosing the right SG for the right task and network, we investigate the SG training sensitivity to task and network complexity. We estimate the task complexity by the number of classes. Thus, if $C_T(\cdot)$ measures task complexity, $C_T(sl\text{-}MNIST) < C_T(SHD) < C_T(PTB)$. We quantify neural complexity as in (Yin et al., 2021), and Tab. 1, by the number of operations performed per layer. In essence, if $C_M(\cdot)$ measures model complexity, then $C_M(LIF) < C_M(ALIF) < C_M(sLSTM)$. To have comparable losses across tasks and networks, we normalize their validation values between

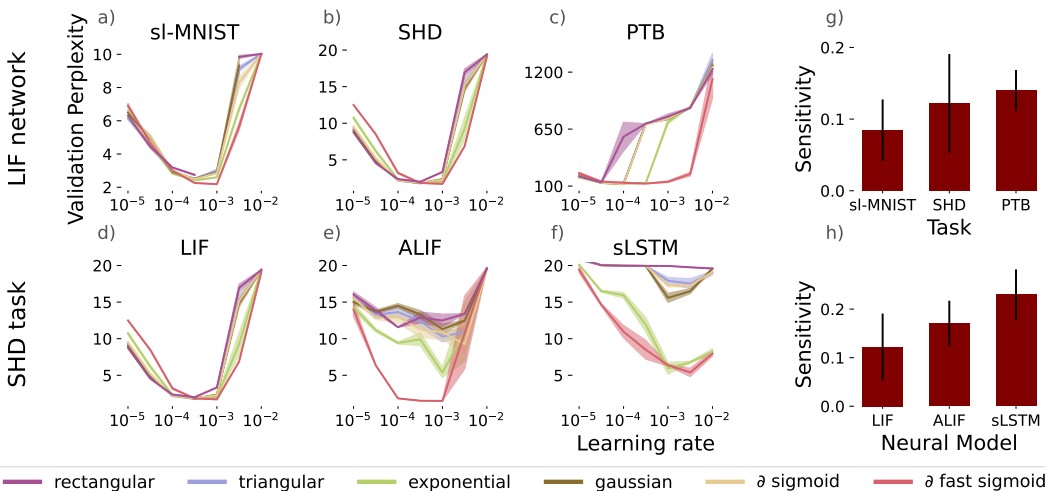

Figure 2: **The derivative of the fast-sigmoid outperforms other SG across tasks and networks**. Grid search over SG shapes, learning rates, tasks and networks. Perplexity is a loss, so, the lower the better. We report lowest validation perplexity after converged training. Panels a-f) show perplexity (y-axis), against learning rate (x-axis). In a-c) we fix the LIF network and change task, while in d-f) we fix the SHD task and change network. Panels g-h) show SG sensitivity (y-axis) against task and neural model (x-axis). Plots b) and d) are repeated for clarity. a-f) Different SG roughly agree on the best learning rate. The best loss is achieved by the ∂ fast-sigmoid, also the most resilient to changes in the learning rate, as shown in (Zenke and Vogels, 2021). g-h) The more complex the task or the network, the more variance in the performance we see across SG choices and learning rates.

0 and 1. For that, we remove the lowest loss achieved by a network in a task for any seed and learning rate, and divide by the distance between the highest and lowest loss. We call the result the *post-training normalized loss*. We call *sensitivity* the standard deviation of the *post-training normalized perplexity* across SG, for each learning rate. We report mean and standard deviation across learning rates.

### 3.1.2 EXPERIMENTS

We see in Fig. 2, that task and network complexity have a measurable effect on the sensitivity of training to the SG choice. We run a grid search over learning rates and SG shapes. The sensitivity to the task is shown in the upper panels, for the LIF network. We see that different SG agree on the optimal learning rate. We also see that the ∂ fast-sigmoid performs well for a wider range of learning rates. The rectangular SG is competitive on some tasks, but fails to learn with most learning rates on PTB. Then we focus on network sensitivity, fixing the SHD task, lower panels. The triangular SG performs similarly to the exponential on the LIF network, while it underperforms on ALIF, and fails on sLSTM. The exponential SG matches the best SG on both the LIF and the sLSTM, but not on the ALIF. All this manifests a strong sensitivity to the SG choice. Surprisingly, the sLSTM lags behind the LIF and ALIF, with a comparable number of parameters. The gating mechanism devised to keep the LSTM representations from exploding exponentially, are not relevant anymore for a Heaviside that cannot explode exponentially, and might have become a computational burden. Fig. 2, g-h), confirm that there is a correlation between task and network complexity, and SG sensitivity.

### 3.2 ORTHOGONAL INITIALIZATION LEADS TO HIGHER ACCURACY

In contrast, SG training seems less sensitive to the weights initialization scheme. Fig. 6 App. C, shows the initialization scheme effect on SG training, for the LIF network on the SHD task. We compare the *Glorot* ($2/Var[W_l] = n_{l-1} + n_l$, (Glorot and Bengio, 2010)) with the *He* ($2/Var[W_l] = n_{l-1}$, (He et al., 2015)) and the *Orthogonal* initialization ($W_{rec}$

and $W_{in}$ as column orthogonal matrices, (Saxe et al., 2014)). We use three sampling distributions: Uniform, Gaussian and BiGamma, Fig. 1. Orthogonal Uniform is not considered since after orthogonalization, the distribution was not Uniform anymore. We use the same initialization for $W_{rec}$ and $W_{in}$. The best SG with *Glorot* Uniform is the $\partial$ fast-sigmoid, while with *Orthogonal* Normal is the exponential. *He* gives the best outliers, and *Orthogonal* Normal gives the best mean accuracy. The BiGamma reduces the result variance. Overall best mean is achieved by the derivative of the fast-sigmoid and by the exponential SG, suggesting similar behavior across initializations.

### 3.3 HIGH INITIAL FIRING RATE CAN PROMOTE GENERALIZATION WITH LOW FINAL FIRING RATE

#### 3.3.1 METHODOLOGY: SPARSITY AND BINARITY ROLES IN GENERALIZATION

In order to propose our theoretical method for SG choice, we want to make sure that high initial firing rates are not pernitious neither for learning nor for final sparsity. This is so, because in the neuromorphic literature training success is judged by (1) training performance and (2) activity sparsity. We show in Sec. 3.3.2, that low initial sparsity can improve generalization in synergy with a sparsity encouraging loss term (SELT). However, the energy gains of spiking networks also come from their binary activity. A matrix-vector multiplication, with a $\mathbb{R}^{m \times n}$ matrix, has an energy cost of $mnE_{MAC}$ for a real vector, and of $mnpE_{AC}$ for a binary vector, where $p$ is the Bernouilli probability of the binary vector, and in our case the neuron firing rate, and $E_{AC}, E_{MAC}$ are the energies of an accumulate and a multiply-accumulate operation (Yin et al., 2021; Hunger, 2005). We quantify the sparsity of a binary vector as $1 - p$. Since MAC are more costly than AC, 31 times on a 45nm complementary metal–oxide–semiconductor (Yin et al., 2021; Horowitz, 2014), we have energy savings with any $p$, e.g., when all neurons fire ($p = 1$) and when they fire half of the time steps ($p = 1/2$). This gain does not depend on the simulation speed, since it compares a spiking and an analogue computation, at the same computation speed.

We measure the Pearson correlation of initial and final firing rate $p_i, p_f$ with test loss after training, in two settings, with and without a SELT. The SELT is a mean squared error between a target firing rate $p_t = 0.01$ and the layer firing rate. To achieve different $p_i$, we pre-train $\boldsymbol{b}_l$ on the dataset of interest, holding the other parameters untrained, using only the SELT without the classification loss. The coefficient to multiply the loss term is chosen to make all losses comparable only when the task is learned, to let the network focus first on the task and then on the sparsity. We therefore chose as the multiplicative factor the minimal training loss achieved without SELT, since the SELT takes values between zero and one. We switch on the SELT gradually during training. The switch starts as zero, and moves linearly to one between $1/5$ and $3/5$ of training. We focus on the $\partial$ fast-sigmoid and the SHD task in the main text, but we show different SG and tasks in App. H.

#### 3.3.2 EXPERIMENTS

We can see in Fig. 3 that with and without a SELT, higher $p_i$ correlates with performance. Correlations are bold when $p$-value $\leq 0.05$. Notice that SELT achieved worse final train loss (not shown). However, the high $p_i$ combined with SELT resulted in better test loss, thus, better generalization. However, this is not consistent across SG shapes, Fig. 7, but is consistent across tasks, Fig. 8 App. H. In fact, the triangular SG prefers low $p_i$ and the exponential SG does not show a clear trend. Incidentally, the lower layer always reaches higher sparsity, across seeds (Fig. 3), SG shapes (Fig. 7) and tasks (Fig. 8).

### 3.4 OUR THEORETICAL METHOD FOR SG CHOICE IMPROVES EXPERIMENTAL PERFORMANCE

Keeping in mind that we can exploit a low initial sparsity as a regularization mechanism, we propose a method for SG design in spiking recurrent networks inspired by FFN initializations (Glorot and Bengio, 2010; He et al., 2015), that keeps stable gradients with time. We propose four conditions, as four hypothesis to test, that result in a SG that depends on the network and the task. We present the mathematical equivalent in each subsection

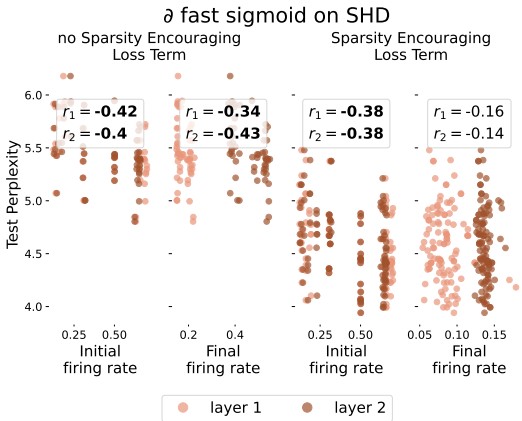

**∂ fast sigmoid on SHD**

Figure 3: **High initial firing rate can promote generalization with low final firing rate.** We use SHD task and the $\partial$ fast-sigmoid SG. Bold correlation means $p$-value $\leq 0.05$. On the two left panels, learning starts from different $p_i$ without a SELT, while on the two right panels a target sparsity is encouraged. In both cases, the initial firing rate correlates with final performance, and a low $p_f$ is achieved successfully using a SELT. Notice as well that the combination of high initial firing rate and sparsity encouragement resulted in better test accuracy than on the two panels on the left, suggesting that both factors acted synergistically as a regularization mechanism.

- I Each neuron has to fire half of the timesteps.
- II Recurrent and input variances should match.
- III Gradients must have equal maxima across time.
- IV Gradients must have equal variance across time.

### 3.4.1 RECURRENT MATRIX MEAN SETS THE FIRING RATE (I)

As previously seen, high $p_i$ can generalize better with a lower $p_f$. Moreover, notice that SG curves reach their highest when the neuron fires, Fig. 1. Thus, if the voltage stays close to firing, the gradient is stronger, which is always so if $Median[v] = 0$ and $Var[v] = 0$. However, $Var[v] = 0$ turns off all higher moments, thus, we only assume $Median[v] = 0$ as the mathematical equivalent of our desiderata. When (I) is applied to a LIF network, see App. E.1, the mean of the recurrent weight matrix fixes $p_i$, further assuming $\overline{w}_{in} = 0$, $\boldsymbol{b} = 0$, the approximation $Mean[v] \approx Median[v]$, and constant $\bar{i}_t$ over time, we find

$$\overline{w}_{rec} = \frac{1}{n_l - 1}\Big(2 - \alpha_{decay}\Big)\vartheta \tag{I}$$

The assumption $Mean[v] \approx Median[v]$, can be justified by noticing that if $v$ is sampled from a unimodal distribution with the first two moments defined, then $|Mean[v] - Median[v]| \leq \sqrt{0.6Var[v]}$ is true (Basu and DasGupta, 1997). Experimentally, we observe always unimodal distributions, that verify $|Mean[v] - Median[v]| \leq \sqrt{c\,Var[v]}$, with $c = 10^{-4}$ for the SHD task, $c = 3 \times 10^{-2}$ for sl-MNIST, and $c = 10^{-3}$ for PTB, with and without (I).

### 3.4.2 RECURRENT MATRIX VARIANCE CAN MAKE RECURRENT AND INPUT CONTRIBUTION TO VOLTAGE COMPARABLE (II)

Also pertaining the forward pass, we want the neuron to be as sensitive to the network history as it is to new input at initialization, when the structure of the task is unknown. We describe it mathematically as $Var[W_{rec}x_{t-1}] = Var[W_{in}z_t]$. In a LIF network, it sets the variance of the recurrent matrix that makes both contributions equal, see App. E.2, further assuming $\mathbb{E}\,x = 1/2$, $\overline{w}_{in} = 0$, and computing $Var[z_t]$ and $\bar{z}_t$ on the train set, we obtain

$$Var[w_{rec}] = 2(Var[z_t] + \bar{z}_t^2)\frac{n_{l-1}}{n_l - 1}Var[w_{in}] - \frac{1}{2}\overline{w}_{rec}^2 \tag{II}$$

### 3.4.3 DAMPENING AND SHARPNESS SET GRADIENT MAXIMUM AND VARIANCE (III, IV)

Instead, to control the backward pass, we want stable gradients with time. We describe mathematically (III) as $Max[\frac{\partial}{\partial\theta}y_t] = Max[\frac{\partial}{\partial\theta}y_{t-1}]$ and (IV) as $Var[\frac{\partial}{\partial\theta}y_t] = Var[\frac{\partial}{\partial\theta}y_{t-1}]$. On a LIF network, they set the dampening and the second moment of the SG that keep the maximum and variance of the gradient stable with time, see App. E.3, E.4. Sharpness and tail-fatness are linked to the SG second moment, see App. E.5. Assuming $\sigma'$ and $\frac{\partial}{\partial\theta}y_{t-1}$ as

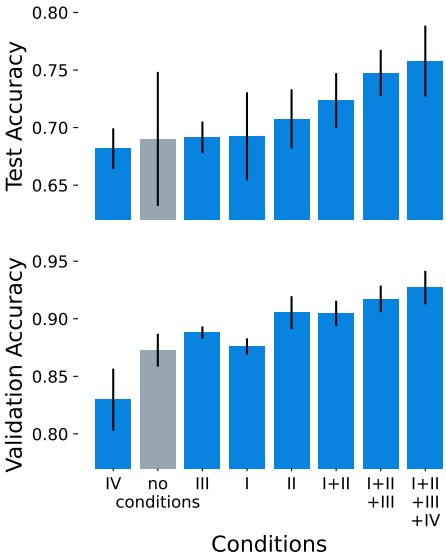

Figure 4: **Our theoretical method for SG choice improves experimental performance.** Simulations are run on the SHD task for a LIF network. We propose 4 conditions to select a SG for a LIF networks. (I) requires spiking half the timesteps, (II) encourages comparable presynaptic and input contribution to voltage, while (III) and (IV) encourage gradient maxima and variance constant with time. The baseline has *no conditions* applied, in gray, with an exponential SG and Glorot Uniform initialization. Lower and upper panels show validation and test accuracies. (II) has the highest impact on its own, but all conditions combined achieve the best result, showing that a theory of SG design can reduce the need of extensive hyper-parameter search.

independent, and zero mean gradients at initialization, we find

$$\gamma = \frac{1}{(n_l - 1)\hat{w}_{rec}}\Big(1 - \alpha_{decay}\Big) \tag{III}$$

$$\overline{\sigma'^2} = \frac{1 - \frac{1}{2}\alpha_{decay}^2}{(n_l - 1)\overline{w^2}_{rec}} \tag{IV}$$

### 3.4.4 Experiments

Fig. 4 shows training results with our conditions for the LIF network on the SHD task, with exponential SG, against the unconditioned baseline. (II) improves accuracy the most when applied on its own, but the best performance is achieved with all conditions together. When all conditions are applied, a LIF network achieves a $92.7 \pm 1.5$ validation and $75.8 \pm 3.1$ test accuracy, compared to $87.3 \pm 1.4$ validation and $69.0 \pm 5.8$ test accuracy without conditions.

### 3.5 The conditions predict best empirical dampening, sharpness and tail-fatness on the LIF network

We compare experimentally the performance of a range of values of dampening, sharpness and tail-fatness and we assess how they compare to the theoretical prediction. Fig. 5 shows the accuracy of the LIF network on the sl-MNIST task. Each SG has its tail decay: inverse quadratic for the $\partial$ fast-sigmoid, no tail for the triangular and rectangular, and exponential decays for the rest. Low dampening and high sharpness are preferred by all SG. Interestingly, the accuracy of the $\partial$ fast-sigmoid degrades less with suboptimal $\gamma, \beta$. The vertical dashed lines are predicted by our theoretical method, condition (III) for the dampening and (IV) for the sharpness of an exponential SG. We observe that they find $\gamma, \beta$ with high experimental accuracies. This supports the claim that to reduce hyper-parameter search of dampening and sharpness is possible. We use our $q$-PseudoSpike SG to study the dependence with the tail-fatness, panel (c) Fig. 5. All tail-fatness values perform reasonably well, with a maximum at $q = 1.56$, smaller than the $q = 2$ of the $\partial$ fast-sigmoid. Interestingly our theoretical solution gives a $q = 1.898 \pm 0.002$, surprisingly close to the experimental optimum.

## 4 Discussion and Conclusions

Surrogate Gradients have reduced the gap between Spiking Neuromorphic Computing and Deep Learning, with the consequent energy efficiency gains. Different SG can achieve similar performance, at the expense of potentially costly hyper-parameter search. Our goal was to

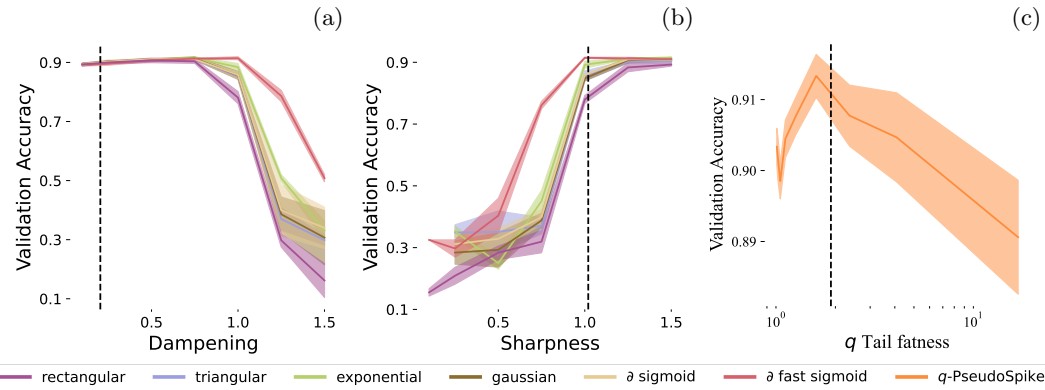

Figure 5: **Low dampening, high sharpness and low tail-fatness, lead to higher accuracy on the LIF network.** Analysis done on the LIF network over the sl-MNIST task. Panel (a) shows performance for different values of dampening, sharpness set to 1, and viceversa on (b). Dampenings higher than 1 worsen performance while the pattern is the opposite for sharpness. Dashed vertical lines are our theoretical prediction for the exponential SG, (III) for the dampening ($\gamma = 0.20 \pm 0.02$) and (IV) for the sharpness ($\beta = 1.02 \pm 0.17$), which agree with the experiments. Panel (c) shows tail-fatness sensitivity of the $q$-PseudoSpike SG, for $\beta = \gamma = 1$. The theoretical prediction gives a close to optimal $q = 1.898 \pm 0.002$, where the best experimentally was $q = 1.56$.

reduce the need of such search in the future, with experimental and theoretical insights. We saw that best SG across networks and tasks was the derivative of the fast-sigmoid, also known as SuperSpike, and the SG sensitivity increase with task and network complexity. Incidentally, we reached spiking state-of-the-art on the PTB task with the triangular SG. Best average over 12 seeds had $122.8 \pm 10.7$ validation and $114.2 \pm 9.2$ test perplexity, and best seed had 117.2 validation and 109.5 test perplexity. Previous spiking SOTA on PTB was 137.7 test perplexity (Woźniak et al., 2020). Then, we saw the *Orthogonal* Normal as the best initialization across SG, on the LIF network and the SHD task, and our BiGamma weight distribution reduced final variance. We saw that for some SG, a high initial firing rate, combined with a sparsity encouraging loss term, can improve generalization.

The literature on optimal SG is growing in activity (Neftci et al., 2019; Zenke and Vogels, 2021; Yin et al., 2021). However, a theoretical framework was needed. We provide a principled method for initialization and SG design, in the form of four conditions. All of them apply to any architecture. We derived the implications on the LIF, and we saw experimentally improved training. The dampening, sharpness and tail-fatness predicted were among the best empirically, Fig. 5. We saw a preference for low dampening, high sharpness and low tail-fatness, making the ideal SG close to a delta, with heavy tails. Also, passing gradients for voltages far from zero, could allow the network learn from outliers. Therefore, our method can help reduce the costly hyper-parameter grid search. Conditions (II), (III) and (IV) are not restricted to spiking neurons. (I) was introduced to find the SG maxima, but all deep learning activations, have their regime of interest around zero. So, (I) is also universal. Even if not stated, (Glorot and Bengio, 2010; He et al., 2015) use (I) in the form $Mean[v_l] = 0$, to determine that $Mean[W_l] = Mean[b_l] = 0$ at initialization. Our theoretical solution applies to convolutional $W_{in}$ and $W_{rec}$. Take $Var[W_{in}] = n_k Var[w_{in}]$ where $n_k$ is the number of presynaptic neurons for each postsynaptic neuron. In our case $n_k$ was equal to $n_{l-1}$. For rD convolutions, $n_k = k^r n_f$, where $k$ is the kernel size, $n_f$ the number of filters and typically the convolution dimension is $r = 1, 2$. Our conditions also apply to different reset methods, App. F. Notice that our conditions could fix $\gamma$, $\beta$, and $q$, which turns them into tools for SG design. This leads to a new theoretical understanding of the roles played by dampening and sharpness. The dampening keeps the maximal gradients stable through time, while the sharpness keeps the gradient variances stable through time. In summary, this work is in response to the call made by (Zenke and Vogels, 2021) for a theory of SG choice, and it is a first step in that direction.

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
