# OpenReview forum: "Surrogate Gradient Design for LIF networks"
_ICLR.cc/2023/Conference — Submitted to ICLR 2023_

### Official Review · Reviewer_fzV9 · 2022-10-24

**Confidence:** 4
**Clarity, Quality, Novelty And Reproducibility:** nothing to mentioned
**Correctness:** 2
**Technical Novelty And Significance:** 2
**Empirical Novelty And Significance:** 2
**Recommendation:** 3

**Strength And Weaknesses:**

Weaknesses:
The weaknesses are detailed above, including:
1. This paper is hard to follow due to the unclear organization and non-standard presentation.
2. Low quality. There lacks apposite definitions or measures of many terminologies, such as low dampening, high sharpness, and low tail-fatness.
3. The assumptions and conclusions of the theoretical investigation is far from clear.

**Summary Of The Paper:**

As claimed by this work, the authors present a framework for designing the best surrogate gradients of SNNs with LIF neurons.

Unfortunately, this paper is hard to follow due to the unclear organization and non-standard presentation. The claimed contributions can be roughly divided into two terms, i.e., summarizing the observed results from experiments and presenting the theoretical understanding of SG design. I cannot demonstrate the correctness of the observed results since there lacks apposite definitions or measures of many terminologies, such as low dampening, high sharpness, and low tail-fatness.

More importantly, this paper provides a theoretical method for SG choice to improve the experimental performance. What are the assumptions and conclusions of this theoretical investigation?

Besides, I would appreciate to have a clear clarification on the experimental details before showing the experimental results, such as shown in Subsection 3.5.5.

**Summary Of The Review:**

Overall, I believe that the presence of this paper is poor, and thus, it is difficult to verify the quality of its work, which would move the paper away from the accepted standard.

---

> ### Author Response · Authors · 2022-11-17
> **Reply to Reviewer 4**
>
>
> Unfortunately, this paper is hard to follow due to the unclear organization and non-standard presentation. The claimed contributions can be roughly divided into two terms, i.e., summarizing the observed results from experiments and presenting the theoretical understanding of SG design. I cannot demonstrate the correctness of the observed results since there lacks apposite definitions or measures of many terminologies, such as low dampening, high sharpness, and low tail-fatness.
>
>     - Sorry to hear that but thanks for the feedback. Dampening and sharpness are defined in the introduction and in section 2.3. Tail-fatness is defined for the q-PseudoSpike in App. D. We reworked the logical coherence of the article to reply to several reviewers that made the same observation, hopefully it should be a better read now.
>
> More importantly, this paper provides a theoretical method for SG choice to improve the experimental performance. What are the assumptions and conclusions of this theoretical investigation?
>
>     - Hopefully it should be more clear now that the text has been reorganized. The conclusion is basically that now the community has access to equations I, II, III and IV, that can be applied to new LIF, and we proved that they result in better performance.
>
> Besides, I would appreciate to have a clear clarification on the experimental details before showing the experimental results, such as shown in Subsection 3.5.5.
>
>     - The experimental setting is similar in all experiments. The only changes are specified in each section, such as in some section we change the initializations, in some the dampening, etc, but the baseline architecture and settings are in section 2.5 and Appendix A!
>
> Strength And Weaknesses:
> Weaknesses: The weaknesses are detailed above, including:
>
> This paper is hard to follow due to the unclear organization and non-standard presentation. (Methodology and Results)
>
>     - We decided to go for this presentation since we felt we had a lot of methodology that was tricky to remember at each phase of the results presentation. The goal was to make it easier for the reader to remember what part of the methodology the results were mentioning. We reworked the logical coherence of the text, and hopefully it should be an easier read now.
>
>
> Summary Of The Review:
> Overall, I believe that the presence of this paper is poor, and thus, it is difficult to verify the quality of its work, which would move the paper away from the accepted standard.
>
>     - Hopefully the new version can change your mind. Thanks a lot for the review anyway!

---

### Official Review · Reviewer_xUW9 · 2022-10-24

**Confidence:** 4
**Correctness:** 3
**Technical Novelty And Significance:** 2
**Empirical Novelty And Significance:** Not applicable
**Recommendation:** 3

**Clarity, Quality, Novelty And Reproducibility:**

The paper is not very clearly written.  It is probably reproducible, but the novelty is not so high.

**Strength And Weaknesses:**

Strengths

The paper aims to facilitate the training of spiking neural networks via surrogate gradients, which remains more challenging and less documented compared to more popular, non-spiking artificial networks.
SNNs are applied to rather diversified tasks involving image recognition, spoken digit recognition and language modelling.
Some findings can lead to helpful, practical solutions for SNN training.

Weaknesses

The overall narrative of the paper needs substantial improvements. A fair number of (sub)sections do not lead smoothly into the next ones. The number of components tested in the study is rather ambitious, which can be okay, but the different findings often feel disconnected, which makes the general direction hard to follow.
For instance, the authors define a spiking version of the LSTM, but the results with it are only shown to be inferior to those obtained with simpler, more commonly used and more biologically plausible LIF and ALIF models. Even if interesting, the inclusion of this spiking LSTM does not seem justified and feels out of the way in terms of the general objectives of the paper.
The use of the different models and data sets does not feel coherent. Some hypotheses are tested on all tasks and models, whereas others for instance only use the LIF on a single task.
The paper feels more like a report than an actual article.

Remarks

The SHD data set only seems to have training and validation splits in its default version. It would be worth explaining what test split you are using.
The spiking LSTM, which is said to be defined in Appendix G, is missing (maybe appendixes are simply not included for this review process, in which case, this remark can be discarded).

**Summary Of The Paper:**

This paper considers different spiking neuron models and explores hyperparameters on multiple tasks in order to facilitate efficient training via the surrogate gradient method. The study focuses primarily on the shape of the surrogate gradient function. They find that low dampening, high sharpness and low tail-flatness generally lead to better performance. More precisely, the fast-sigmoid derivative appears as the best choice of surrogate gradient function. Secondly, they perform a study of different weight initialisations, which reveals that “He” gives the best outliers, “orthogonal normal” the best mean accuracy and “bigamma” the lowest variance. Thirdly, they demonstrate that using high initial firing rates combined with a sparsity encouraging loss term induces better generalisation. Finally, they investigate various conditions to ensure stable gradients and show that these can also lead to better performances.

**Summary Of The Review:**

The authors test a variety of techniques, albeit somewhat inconsistently.  As a review the paper could be useful, but overall it comes across as rather rushed and incomplete.

---

> ### Author Response · Authors · 2022-11-17
> **Reply to Reviewer 3**
>
>     - Thanks a lot for the feedback!
>
>
> Weaknesses
>
> The overall narrative of the paper needs substantial improvements. A fair number of (sub)sections do not lead smoothly into the next ones. The number of components tested in the study is rather ambitious, which can be okay, but the different findings often feel disconnected, which makes the general direction hard to follow.
>
>     - Following your suggestion and similar impressions from other reviewers, we made major changes in the coherence of the article. Hopefully it became more readable!
>
> For instance, the authors define a spiking version of the LSTM, but the results with it are only shown to be inferior to those obtained with simpler, more commonly used and more biologically plausible LIF and ALIF models. Even if interesting, the inclusion of this spiking LSTM does not seem justified and feels out of the way in terms of the general objectives of the paper.
>
>     - We were interested in augmenting the neuron complexity without changing the number of parameters, not in the spiking LSTM itself, and at the same time, we wanted to use a model that did not require too many justifications. That's why we didn't want to elaborate more on it! I tried to put less emphasis on it in the current version, and justify better that we want to show the sensitivity to complexity.
>
> The use of the different models and datasets does not feel coherent. Some hypotheses are tested on all tasks and models, whereas others for instance only use the LIF on a single task. The paper feels more like a report than an actual article.
>
>     - Sorry to hear that, but thanks for pointing out. We reworked the logical coherence of the article, and hopefully it should be an easier read in the current version. Given that there are 7 SG to test, it becomes quickly unfeasible to test all initialization schemes on all tasks, all dampenings and all sharpness values for example, that's why some selections had to be done.
>
> Remarks
>
> The SHD data set only seems to have training and validation splits in its default version. It would be worth explaining what test split you are using. The spiking LSTM, which is said to be defined in Appendix G, is missing (maybe appendixes are simply not included for this review process, in which case, this remark can be discarded).
>
>     - The Appendix is in the Supplementary Material .zip, since for the paper there is a 9 pages limit. The dataset samples for train, validation and test, are specified in the Appendix as well! But for SHD the validation set is 833 samples taken from the train set.
>
> Clarity, Quality, Novelty And Reproducibility:
> The paper is not very clearly written. It is probably reproducible, but the novelty is not so high.
>
>     - We rewrote the whole article to reply to this fair criticism with an improved logical coherence. On the novelty, we think that using the variance stability method to design the SG has never been done before! Thanks a lot anyway for the feedback.

---

### Official Review · Reviewer_HAej · 2022-10-25

**Confidence:** 5
**Correctness:** 3
**Technical Novelty And Significance:** 2
**Empirical Novelty And Significance:** 2
**Recommendation:** 3

**Clarity, Quality, Novelty And Reproducibility:**

The paper seem to be a rush re-wrap from its long preprint. Thus the writing and readability are not sufficiently good.

**Strength And Weaknesses:**

Strengths:
1.	The authors provide empirical results on how to select SGs and initial methods to enhance the final performance.
2.	The authors designed a new kind of SG that is suitable for all the datasets they used.
Weaknesses:
1.	Lack of comparison with other existing work.
2.	This paper only provides experiments on very simple datasets and network structures. More complex cases need to be included.
3.	The logic of this paper is confusing, e.g. many important figures are placed in the appendix, making it difficult for the readers to read.
4.	Their main findings are obtained through some experiments with grid search while lacking logic proof. So the article's novelty and contribution are modest.

Questions:
1. In Sec 3.1.2, what is the complete model structure for testing different SGs?
2. 1.The datasets used in the experiments seem to be relatively small datasets. How does the experiment perform on larger datasets? I think the baseline of this work should be compared with the existing work.
3. 1.The SG used doesn't guarantee $\int_{-\infty }^{+\infty }f(x) = 1$. Maybe that's why the other SGs don't perform well?
4. 1.In sec 3.4.1.Is there a missing equation before "where p is the firing rate"?
5. 1.Fig 5. If the baseline is replaced with other initialization methods, will the results change?


**Summary Of The Paper:**

This article aims to experimentally and theoretically define the best SG across different stress tests to reduce the future need for grid search and proposes techniques to improve the performance of SNN training from multiple perspectives, such as the dampening of the SG and the normalization method.

**Summary Of The Review:**

Overall, it adds an interesting paragraph to the training of SNN with LIF but the performance is not convincing.

---

> ### Author Response · Authors · 2022-11-17
> **Reply to Reviewer 2**
>
> Thanks for the review and commenting on the strengths!
>
>   1. Lack of comparison with other existing work.
>
>     - We wanted to avoid taking results from others in conditions that were not under control and therefore hard to compare, so, we took SG shapes proposed in other works, and compared them in a controlled setting. However, we mention that we achieve spiking networks SOTA in PTB in the Discussion.
>
>   2. This paper only provides experiments on very simple datasets and network structures. More complex cases need to be included.
>
>     - It is true that the architectures and tasks considered in Neuromorphic Computing can give the impression of being very simple for people in Machine Learning. However, it is due to the fact that the binary activity constrains performance significantly, so it is much harder to balance performance and energy efficiency in harder tasks and architecture. Moreover, I only have limited resources at my disposal.
>
>   3. The logic of this paper is confusing, e.g. many important figures are placed in the appendix, making it difficult for the readers to read.
>
>     - Thanks a lot for the feedback. We have put special emphasis on reworking the coherence of the article for this rebuttal, and hopefully the common thread should be clearer now, to reply to your feedback and to the feedback of other reviewers. It is probably what I find hardest of writing in English. Also, I tried to show in the main text only the images that were sufficient to portray the message, while leaving for the Appendix all those that seemed redundant but still important for the curious reader and for completeness.
>
>   4. Their main findings are obtained through some experiments with grid search while lacking logic proof. So the article's novelty and contribution are modest.
>
>     - In the new version I tried to clean the logical coherence of the article. We think that a standardized comparison of different SG was necessary. We also think that using Glorot/He representation theory to design an approximate gradient is quite novel, since it has been always used to set the initialization hyperparameters for the weights, not the shape of the gradient.
>
> Questions:
>
>   1. In Sec 3.1.2, what is the complete model structure for testing different SGs?
>
>     - The network is specified in section 2.5 and 2.2, with extra details in Appendix A.
>
>   2. The datasets used in the experiments seem to be relatively small datasets. How does the experiment perform on larger datasets? I think the baseline of this work should be compared with the existing work.
>
>     - As mentioned above, unfortunately, spiking networks cannot compete in harder tasks, so research still has to focus on simpler ones.
>
>   3.The SG used doesn't guarantee x. Maybe that's why the other SGs don't perform well?
>
>     - Sorry about the misunderstanding. The SG used all satisfy that condition, but it was stated in Appendix D. It used to be in the main text, but to make the content fit to the 9 pages limit, I thought it was information that could be kept in the Appendix. I placed it again in the main text for the new version following your observation.
>
>   4.In sec 3.4.1.Is there a missing equation before "where p is the firing rate"?
>
>     - At least it's missing a better connection with the text that preceeds. I improved the description following your advice.
>
>   5. Fig 5. If the baseline is replaced with other initialization methods, will the results change?
>
>     - It is possible to change, but, we show in appendix C that across different types of initializations, the derivative of the fast sigmoid and the exponential SG seem to consistently outperform the rest. Therefore, it seems to be more robust to the initialization choice.
>
> Overall, it adds an interesting paragraph to the training of SNN with LIF but the performance is not convincing.
>
>     - Thanks a lot for the positive and constructive feedback. We put emphasis on reworking the logical coherence of the text. I hope it has become a better read!

---

> > ### Comment · Reviewer_HAej · 2022-11-20
> > **Response to the rebuttal**
> >
> > Thank you for your response to my questions.
> >
> > In terms of the rebuttal content, as the authors mentioned, it is true that single SNN neurons may be short in representation. This is why you need to design some complementary or special mechanism to improve the representation power here as well as keep the biological efficiency. And this is why a proposed method needs additional examination on the relatively complex dataset to show its efficacy.  Regarding writing, it is not a problem of language but the logic of organizing materials. You may look for help from some senior people in your institute for help.
> >
> > Additionally, as a PI myself, I felt a bit sympathetic for the leading author who appeared to be a student lacking sufficient instruction from his mentor. However, based on the quality of the manuscript and rebuttal, I cannot increase my rating score here. Hope the authors can get better in the future.

---

### Official Review · Reviewer_WUfr · 2022-10-28

**Confidence:** 2
**Correctness:** 3
**Technical Novelty And Significance:** 2
**Empirical Novelty And Significance:** 2
**Recommendation:** 5

**Clarity, Quality, Novelty And Reproducibility:**

The authors of this work do not promise any reproducibility in text.



**Strength And Weaknesses:**

Strength: 1. Training details are explicit, including network architecture and hyper-parameters. 2. Conclusion from the experiments is helpful in choosing surrogate gradient.

Weaknesses: It doesn’t seem reasonable to make the neuron as sensitive to the network history as to new input in section 3.5.2, since the network history is an accumulation of knowlegde learnt from past so it contains more knowledge than new input.

Minor typos:

Sec2, line 3 "y_L the network output" -> "y_L is the network output" or "y_L means the network output"

**Summary Of The Paper:**

This work finds it unclear to determine which surrogate gradient to apply in different tasks or networks in SNN. Seeking to solve this problem, the authors have done some experiments with different surrogate gradients across tasks and networks, and come to the conclusion that the derivative of fast sigmoid outperforms other chosen surrogate gradients. Besides that, the authors have also done research on how different characteristics of surrogate gradients and initialization affect surrogate gradient learning and found that low dampening, high sharpness, low tail-fatness, orthogonal initialization, and high initial firing rate could help improve the network’s performance. Lastly, the authors provide a theoretical solution based on bounding representations to find a surrogate gradient and initialization that could improve accuracy.

**Summary Of The Review:**

The discussion on how to choose SG in SNN is meaningful and missed before.

---

> ### Author Response · Authors · 2022-11-17
> **Reply to Reviewer 1**
>
> Thanks a lot for the review and glad to hear that you found training details explicit and conclusions helpful to choose an SG!
>
> On the weaknesses, it is true that it can be considered a strong choice to make to have a neuron as sensitive to the network history as to new input. However there is a dilemma, since there's two properties that we want our system to have when we don't know which features are going to be important to solve a task: (1) we want the network to be stable and to be able to recall information far in the past, (2) we want the network to be reactive to new information. We thought that a good balance to set at initialization was to give the same importance to both behaviors at initialization and our experimental results in Fig. 5 confirm that it was a good inductive bias choice, but for sure it seems as a question that has potential to be investigated in deeper detail in future works.
>
> To reply to all the reviewers we undertook some substantial rewriting of the paper, to improve the coherence. I hope it can improve your impression of it, and thanks again for the review!

---

### Decision · Program_Chairs · 2023-01-20

**Decision:**

Reject

**Justification For Why Not Higher Score:**

Based on the reviewer comments, this paper is not sufficiently well-written for acceptance at ICLR.

**Justification For Why Not Lower Score:**

N/A

**Metareview: Summary, Strengths And Weaknesses:**

This paper explores the use of surrogate gradients in spiking neural networks. It examines how different shapes of surrogate gradient affect performance and explores a theoretical construction of a good initialization for surrogate gradient parameters that reduce the burden of hyperparameter searches.

The strength of this paper is that it is exploring an important question and could potentially be very useful for researchers working on spiking neural networks.

The weakness of this paper, agreed by all reviewers, is a significant lack of clarity, and in turn, reproducibility.

The authors responded to the reviewers concerns, but it was not enough to change the low scores (average score of 3.5). Based on this, a decision of reject was reached.

**Summary Of Ac-Reviewer Meeting:**

N/A